# Determinants of Adherence to a “GRADIOR” Computer-Based Cognitive Training Program in People with Mild Cognitive Impairment (MCI) and Mild Dementia

**DOI:** 10.3390/jcm11061714

**Published:** 2022-03-19

**Authors:** Angie A. Diaz Baquero, María V. Perea Bartolomé, José Miguel Toribio-Guzmán, Fernando Martínez-Abad, Esther Parra Vidales, Yolanda Bueno Aguado, Henriëtte G. van der Roest, Manuel A. Franco-Martín

**Affiliations:** 1Institute of Biomedical Research of Salamanca, University of Salamanca, 37001 Salamanca, Spain; mfm@intras.es; 2Department of Research and Development, INTRAS Foundation, 49001 Zamora, Spain; jmtg@intras.es; 3Basic Psychology, Psychobiology and Methodology Department, Salamanca University, 37001 Salamanca, Spain; vperea@usal.es; 4University Institute of Educational Sciences, University of Salamanca, 37008 Salamanca, Spain; fma@usal.es; 5IBIP Center for Clinical Care in Mental Health and Aging, INTRAS Foundation, 49024 Zamora, Spain; epv@intras.es; 6Gradior Department and Cognitive Research, INTRAS Foundation, 47016 Valladolid, Spain; yba@intras.es; 7Department on Aging, Netherlands Institute of Mental Health and Addiction (TrimbosI Institute), 1013 GM Utrecht, The Netherlands; hroest@trimbos.nl; 8Psychiatric Department, Rio Hortega University Hospital, 47012 Valladolid, Spain; 9Psychiatric Department, Zamora Healthcare Complex, 49071 Zamora, Spain

**Keywords:** dementia, rehabilitation, software, computer-based, cognition, psychology

## Abstract

Background: Computer-based programs have been implemented from a psychosocial approach for the care of people with dementia (PwD). However, several factors may determine adherence of older PwD to this type of treatment. The aim of this paper was to identify the sociodemographic, cognitive, psychological, and physical-health determinants that helped predict adherence or not to a “GRADIOR” computerized cognitive training (CCT) program in people with mild cognitive impairment (MCI) and mild dementia. Method: This study was part of a randomized clinical trial (RCT) (ISRCTN: 15742788). However, this study will only focus on the experimental group (*n* = 43) included in the RCT. This group was divided into adherent people (compliance: ≥60% of the sessions and persistence in treatment up to 4 months) and non-adherent. The participants were 60–90 age and diagnosed with MCI and mild dementia. We selected from the evaluation protocol for the RCT, tests that evaluated cognitive aspects (memory and executive functioning), psychological and physical health. The CCT with GRADIOR consisted of attending 2–3 weekly sessions for 4 months with a duration of 30 min *Data analysis:* Phi and Biserial-point correlations, a multiple logical regression analysis was obtained to find the adherence model and U Mann–Whitney was used. Results: The adherence model was made up of the Digit Symbol and Arithmetic of Wechsler Adult Intelligence Scale (WAIS-III) and Lexical Verbal Fluency (LVF) -R tests. This model had 90% sensitivity, 50% specificity and 75% precision. The goodness-of-fit *p*-value of the model was 0.02. Conclusions: good executive functioning in attention, working memory (WM), phonological verbal fluency and cognitive flexibility predicted a greater probability that a person would be adherent.

## 1. Introduction

The World Alzheimer Report 2019 estimated that there are more than 50 million people with dementia (PwD) in the world, a number that will increase to 152 million by 2050 [1]. In recent years, various investigations have been developed on the effectiveness of computer-based cognitive training (CCT) programs to help delay decline and maintain cognitive status in people with mild cognitive impairment (MCI) and mild dementia [2,3]. However, it is not only important to study the effectiveness, but it is also necessary to take into consideration the rate and therefore, the possible characteristics that determine adherence to CCT program.

Adherence has been mostly studied in association with the use of drugs [4], there being few studies about adherence to CCT programs in people with MCI and dementia [5]. The WHO defines adherence as the degree to which a person’s behavior with respect to a treatment corresponds to the recommendations provided by a healthcare professional [6]. Therapeutic compliance has been used as a synonym for adherence, this refers to the degree to which a patient acts according to a therapeutic regimen [7]. That is, the frequency, duration, and latency of a specific treatment. Recently, the literature emphasized persistence to help complement the definition of adherence, terms often used interchangeably, but differing. Persistence refers to the time of treatment from its beginning to its end; in which, there could be a “grace period”. That is, a permitted time interval in which the person temporarily suspends the treatment, but retakes it until its end [8]. Spanish Society of Pharmacy, Clinic, Family and Community (SEFAC) proposes that for a patient to be completely adherent, he/she must be compliant and persistent [9]. However, there will be cases where a person may be compliant, but not necessarily persistent, vice versa or neither [10].

The WHO pointed out the lack of adherence as a public health problem and proposed some factors that could determine it, such as: socioeconomic level, the person, the therapy, different conditions, the health system, and care team [9]. Our research will particularly focus on investigating the determinants of adherence associated with the person.

So, adherence can be predicted and explained from a cognitive, psychological, social, neurocognitive, and even technological point of view. Scase et al. [11] pointed out the mild level of deterioration, social interaction, and the availability of technological support to explain the adherence of a group of people with MCI to computer-based gamified environments. Evers et al. [12] mentioned that poor baseline performance in memory, attention, and semantic verbal fluency (SVF) tests helped to predict greater adherence in older women.

Park et al. [13] tried to associate adherence to virtual reality (VR) training program with the improvement of cognitive functioning. Turunen et al. [14] mentioned good memory performance as one of the variables that helped to predict the adherence to a CCT program in adults at risk of dementia. Han et al. [15] found a correlation between the improvement of memory skills and adherence to The Ubiquitous Spaced Retrieval-based Memory Advancement and Rehabilitation Training (USMART) in people with MCI. On the other hand, when older adults are aware of their cognitive decline, specifically when it affects their executive functioning, there was an increase in their adherence [16]. Similarly, de Wit et al. [17] mentioned that the compensatory strategies offered by a memory support system training in people with MCI could influence adherence. Certain psychological variables such as positive expectations at the beginning of a CCT program were found among the main factors that help predict adherence [14]. On the other hand, it seems that people with cognitive impairment spent more time on web-based CT sessions than people with depressive symptoms or other psychiatric disorders [18].

Moreover, earlier studies have already remarked the difficulties in finding features linked to low adherence to the intervention based on individual Cognitive Stimulation Therapy (iCST) [19]. Consequently, the objective of this study was to identify the sociodemographic, cognitive, psychological, and physical-health determinants that helped predict adherence (therapeutic compliance and persistence) or not to a “GRADIOR” CCT program in people with MCI and mild dementia.

## 2. Materials and Methods

### 2.1. Study Design

This study was part of a multicenter simple-blind, randomized clinical trial (RCT) on the effectiveness of the GRADIOR cognitive rehabilitation program in people with MCI and mild dementia [20,21]. This trial was registered at isrctn.com (ISRCTN: 15742788) [22] and was approved by the Drug Research Ethics Committee of the Zamora Health Area (Number: 387-E.C). The recruitment period began in June 2018 and lasted until December 2019. The protocol of this RCT had variations, the main one being related to the design because the ehcoBUTLER platform was excluded. This was not completed on time when the RCT started, which led to modifications in the design with respect to the number of parallel groups, the sample size, and the type of randomization.

The participants included in this RCT were randomly assigned (1:2) to the control or experimental group. Those in the control group (CG) maintained their daily activities, remaining on the waiting list, and those in the experimental group (EG) attended CCT sessions using the GRADIOR program for 4 months. However, the present study focused on the EG. From which, two groups were formed: adherent (compliance-persistent) and non-adherent (those who did not meet any or none of the conditions).

### 2.2. Participants

The study sample included 43 participants aged 60 to 86, and 72.1% had a basic primary educational level. Participants were selected from day centers, memory clinics and hospitals in the Spanish regions of Castile and León and Galicia. *The inclusion criteria* were as follows: (1) clinical diagnosis of MCI according to Petersen’s criteria [23] and mild dementia according to the diagnostic criteria of the Diagnostic and Statistical Manual of Mental Disorders (DSM-V) [24] (this diagnostic was carried out by a psychogeriatrician and neurologist); the types of MCI included were amnestic, and for mild dementia were Alzheimer’s disease, vascular dementia, mixed dementia, and frontotemporal dementia; (2) score on the Yesavage Geriatric Depression Scale (GDS) ≤ 5; (3) voluntary participation of each people; (4) participation of a reference caregiver; 5) speaking and understanding Spanish. An additional criterion was the Mini-Mental State Exam (MMSE) scores, the cut-off point for MCI was ≤27 and for mild dementia was 20 ≥ X ≤ 25. MMSE scores were adjusted according to age and educational level of each people [25].

*The exclusion criteria* were: (1) severe physical comorbidity; (2) severe sensory alterations (auditory or visual) indicated after clinical evaluation by the psychogeriatrician or neurologist; (3) clinically proven psychopathological disorders (depression, anxiety, bipolar disorder, psychosis); (4) neurological disorders (Huntington’s disease, stroke, Parkinson’s disease, dementia with Lewy bodies); (5) history of substance use (e.g., alcohol, tobacco).

To make up the group of adherents, the conditions of compliance and persistence were considered. Regarding therapeutic compliance, people who attended at least 66% of the sessions of the maximum number of sessions, for 30–40 min per session. Additionally, to consider a person persistent or not, the person had to attend weekly for 16 weeks (4 months) of intervention, and therefore they should not exceed the only “grace period” or allowed interval of absence from the CCT of two continuous weeks. If the person met only one or neither of the two conditions, they were considered non-adherent.

### 2.3. Neuropsychological Assessment

Possible predictor variables of adherence were associated with the baseline of RCT. We constructed an evaluation protocol for the RCT, which included several scales that evaluated different aspects. However, we selected the variables for our study, considering the literature and our objectives. Sociodemographic aspects such as age, sex, educational level and years of education were assessed. Appendix A describes each of the test used to specifically measure global cognitive performance, memory and EF: MMSE [26], Memory of words, Word recognition and Total of Alzheimer’s Disease Assessment Scale—Cognitive Sub-scale (ADASCog) [27], Trail Making Test (TMT) forms A–B [28], Digits, Arithmetic and Digit Symbol of Wechsler Adult Intelligence Scale (WAIS-III) [29], Visual Recognition of the Rivermead Behavior Memory Test (RBMT) [30], Visual Reasoning of the Cambridge Cognition Examination (CAMCOG) [31] and Verbal Fluency (Semantic and Lexical) [32].

Affective state, motivation and expectations were also evaluated as part of the psychological dimension (Appendix A). The affective component was evaluated using the GDS [33]. Motivation was valued using the following question: “Do you need someone to encourage you to attend the workshop?” The other questions included in the questionnaire were associated with the following expectations: (1) memory improvement; (2) improvement in quality of life; (3) spending free time in a pleasant way; (4) meeting new people at the workshop. Data on the use of technologies were collected based on the question “Do you usually use any technology?”

Finally, we used the EuroQol (EQ-5D-5L) test [34], which assesses the patients’ perception about the physical-health dimension: mobility, self-care, daily activities, pain-discomfort and anxiety-depression, and patients’ perception regarding their current health condition (Appendix A).

### 2.4. Computer-Based Cognitive Training (CCT) and Adherence

GRADIOR is a computer-based cognitive rehabilitation program and allows CCT of different cognitive functions that present a deficit or deterioration based on different etiologies, including dementia [35]. It includes a series of exercises associated with orientation, memory, attention, perception, executive functioning, reasoning, and calculation (Figure 1). Each of these cognitive modalities (cognitive functions) includes various sub-modalities (cognitive processes) to customize CCT to the user’s cognitive profile [36]. The RCT was based on a cognitive training plan designed for people with MCI and mild dementia, independently (Table 1). This meant that the type of exercises was a function of diagnosis. However, the plan was personalized for each patient according to his/her cognitive level in each exercise. Participants had to attend two or three weekly sessions (this was determined for each center), each lasting 30–40 min. The interface of GRADIOR uses an intuitive touchscreen system and meets usability standards [37,38], providing good user experience [39] to adapt to the needs of older people.

To calculate the adherence rate, the number of sessions attended by each person at the CCT during the 4 months was divided by the maximum number of sessions attended, the result was multiplied by one hundred, except for participants who died or dropped out because of medical reasons, for whom the adherence rate was calculated only until the time of drop-out. We consider the cut-off point of 66% for the rate of adherence or therapeutic compliance according to the literature [5,40,41]. Like compliance, persistence or not was considered as a dichotomous variable [42] and was measured taking into account whether or not the person completed the 4 months of intervention, considering the grace period.

### 2.5. Statistical Analyses

The statistical analysis was performed with the Software Statistical Package for the Social Sciences (SPSS) [43]. We used the punctual biserial correlation to find the degree of association between the dichotomous and dependent variable (adherent and non-adherent) and the quantitative and independent variables (socio-demographic, cognitive, psychological and physical-health). Additionally, the Phi correlation to find the association between dichotomous variables.

The independent variables that were significantly correlated with the dependent variable were taken into consideration as possible predictor variables of adherence and, therefore, were introduced in the analysis with Multiple Logistic Regression. This was used to identify the IVs that helped to predict adherence or not to CCT. We used this analysis because the dependent variable was dichotomous and we had several independent variables, some metric and some qualitative.

Then, we have used a non-parametric analysis for two independent samples (Mann–Whitney) due to the size of the sample and because not all variables followed a normal distribution (Shapiro–Wilk). This analysis allowed us (1) to compare the performance between adherent and non-adherent group, (2) to investigate if there were significant differences between people with MCI and mild dementia in relation to each group (adherent and non-adherent) and, (3) to evaluate if there were significant differences between MCI-adherent vs. MCI-non-adherent and mild dementia-adherent vs. mild dementia-non-adherent with respect to the variables (Digit Symbol and Arithmetic of WAIS-III and LVF-R) that made up the adherence model.

## 3. Results

### 3.1. Participant Characteristics

In total, 140 people were contacted to enter the study, and 47 (33.6%) were excluded due to the following reasons: they did not meet the criteria (*n* = 22) and did not want to participate (*n* = 25). Additionally, four (2.9%) were not randomized due to the onset of COVID-19. A total of 89 (63.6%) people were randomized. Of these, 57 were assigned to the EG. However, only 75.4% (*n* = 43) participants managed to initiate the intervention and 24.6% (*n* = 14) did not start any session due to the start of COVID-19 (Figure 2).

The EG participants were classified into two groups (adherent and non-adherent). Twenty-seven (62.8%) made up the group of adherents. The mean age of this group was 73.6 ± 6.0, and 40.7% were men. The mean years of education were 9.6 ± 2.8. Sixteen (59.3%) participants were diagnosed with MCI and 40.7% (*n* = 11) with mild dementia. The mean adherence rate of this group was 83.3 ± 8.6. The non-adherent group (*n* = 16) had a mean of 76.1 ± 7.5 for age and 8.4 ± 1.1 for years of education. Of this group, 37.5% (*n* = 6) were men, 68.8% (*n* = 11) were diagnosed with MCI and 31.2% (*n* = 5) with mild dementia. Additionally, the mean adherence rate was 59.2 ± 16.1. However, there were no significant differences between the two groups (adherent and non-adherent) with respect to age, sex and years of education (Table 2). Of the non-adherent participants, 56% (*n* = 9) did not comply with 66% of the CCT sessions, 19% (*n* = 3) were not persistent or dropped-out during the training period (4 months), and 25% (*n* = 4) did not meet any of the above conditions (Figure 2). 

### 3.2. Adherence Prediction Model to CCT Program

The variables that were directly correlated with the adherent group to the CCT were: Digit Symbol of WAIS-III (*p* = 0.02), Arithmetic of WAIS-III (*p* = 0.02) and lexical verbal fluency (LVF)-R (*p* = 0.03).

These variables were significantly correlated with the dependent variable and were included in a multiple logistic regression model. The final model proposed that the predictors of adherence were associated with performance in Digit Symbol of WAIS-III (OR: 1.06; 95 CI: 0.77–1.47), Arithmetic of WAIS-III (OR: 1.22; 95 CI: 0.90–1.65) and LVF-R (OR: 1.22; 95 CI: 0.91–1.62) (Table 3). Therefore, an increase in performance in these tests predicts an increase in the probability that a person is adherent (compliant and persistent). The adherent group obtained a better performance with moderate-high effect sizes than the non-adherent group in Digit Symbol of WAIS-III (*p* = 0.016; *r_b_* = 0–444), Arithmetic of WAIS-III (*p* = 0.022; *r_b_* = 0.419) and LVF-R (*p* = 0.014; d-cohen = 0.812) (Table 2). The goodness-of-fit *p*-value of the model was 0.02, with a McFadden R-squared value of 0.174 (Table 3). This model has 90% sensitivity, 50% specificity and 75% precision to correctly identify the adherence group (compliance-persistence). Of the 16 people who were not adherent, the model would only be able to correctly predict 8 subjects.

Regarding the adherent group, there were significant differences with moderate effect sizes between the group of people with MCI and mild dementia with respect to performance in Digit Symbol of WAIS-III (*p* = 0.010; *r_b_* = −0–591), Arithmetic of WAIS-III (*p* = 0.005; *r_b_* = −0.642) and LVF-R (*p* = 0.030; *r_b_* = −0.500). Better performance was seen in people with MCI compared to people with mild dementia (Table 4).

On the other hand, we wanted to compare people with MCI-adherents with MCI-non-adherents and the same for people with dementia. Regarding people with MCI, there were low effect sizes between adherents and non-adherents in relation to performance in Digit Symbol and Arithmetic (*r_b_* = −0.358). We observed a better performance of the MCI-adherent group compared to the MCI-non-adherence group. While in the dementia group, there were low–medium effect sizes (*r_b_* = −0.417) for Arithmetic and LVF-R. We found that the mild dementia-adherent group performed better than the mild dementia-non-adherent group (Table 5).

## 4. Discussion

It is well known that the recruitment and involvement phase of CCT is the most difficult, due to therapeutic nihilism among families and often general practitioners (GPs), and because of patients’ fear of dealing with something new [44]. However, difficulties do not end at the beginning, as patients require motivation to continue the CCT program. In all these stages, we can identify several features that are intrinsic to the person and could intervene and predict adherence to a CCT program. Our purpose was to identify the sociodemographic, cognitive, psychological, and physical-health determinants that helped predict adherence or not to a “GRADIOR” CCT program in people with MCI and mild dementia. It can also be used as guidance to personalize the intervention so that patients might gain more benefits and to improve the sustainability of the care system and take care about the risks of drop-out in specific patients.

### 4.1. Socio-Demographic Variables and Adherence

We did not find age to be a predictor of adherence to CCT, contrary to some studies such as Maseda et al. [45]. Even so, it is difficult to support that age is the only factor that influences adherence itself, but it is probably associated with other problems or capacities that affect adherence. Indeed, age may also be associated with greater cognitive impairment.

Another sociodemographic factor considered was years of education, and we formally hypothesized that it was going to influence on adherence. However, we did not find any relationship between these variables [14]. In contrast, some studies reported educational level as a predictor of adherence [46,47]. We cannot establish any formal recommendations in this regard since different research conclusions are reached. We consider that future studies should assess this aspect more thoroughly to define its relevance. Moreover, it will probably be necessary to combine educational level with other intellectual activities as reading, work performance, hobbies, etc., to identify the main feature involved in adherence.

As for the variables sex and diagnosis, these were not predictors of adherence. However, it is likely that our results were influenced by the greater number of women and people with MCI that made up our sample. Therefore, it would be interesting for future studies to include more balanced groups in terms of sex and diagnosis to determine their level of influence on adherence.

### 4.2. Cognitive Profile and Adherence

Different studies support the presence of executive dysfunction associated with alterations in response inhibition and cognitive flexibility in people with MCI [48] and deficits in working memory (WM) [49], inhibition process, sensitivity to interference [50], flexibility and reasoning [51] in people with Alzheimer’s. In this order of ideas, our findings indicated how some executive functions (EF) such as attention, WM, numerical reasoning, and verbal fluency, evaluated by Arithmetic, Digit Symbol and LVF-R test are involved in adherence to a CCT program such as GRADIOR. Additionally, our previous study on the effectiveness of GRADIOR indicated a trend of improvement in these cognitive processes after 4 months of intervention in the EG [21].

Anderson-Hanley, Arciero, Barcelos, Nimon, Rocha, Thurin and Maloney [16] pointed out EF as a motivator of adherence. In turn, poor executive functioning has been identified as an important factor that helps negatively predict the prognosis of dementia [52].

In particular, the role of WM as an adherence factor has been pointed out in studies on the improvement of this cognitive function (WM) after CCT sessions [53,54], while other studies have associated adherence with memory, the latter from a more general level [14,15]. Other studies have pointed out the association between adherence and delayed recall [55]. The functioning of the WM depends on the process of attention for the performance of the tasks associated with the Arithmetic and Digit Symbol scales. In our study, the WM sub-processes involved in performing these tasks were particularly associated with (1) maintaining information for a short time, which allowed for the (2) using, manipulating, or processing of this information. In some cases, this allowed for the activation of more complex cognitive functions such as numerical reasoning (Arithmetic).

It is probable that the attention, maintenance and manipulating of information associated with WM and the phonological verbal fluency are more directly linked to adherence. In fact, there was a significantly better performance at the beginning of the CCT by the adherent group compared to the non-adherent group in these functions. Likewise, when we simplify the comparison by clinical groups, we find that the MCI/dementia adherent group performed better for these functions compared to the MCI/dementia non-adherent group. In accordance with our findings, we suggest paying more attention to those people with a lower cognitive performance in these tests and, therefore, a greater deterioration in these EF, because they may not be as adherent to a CCT program. So, the development of strategies to link and maintain these people in a CCT program will take more effort, from conducting an adequate neuropsychological evaluation to determining the presence and severity of deficits to planning a personalized CCT intervention plan.

### 4.3. Physical-Health Variables and Adherence

Our findings did not mention any physical-health variable that was able of predicting a person’s adherence to a CCT program such as GRADIOR. Most participants agreed that they did not have mobility problems, self-care, ADL, pain and discomfort, anxiety, and depression. Possibly, if we had found greater problems in these dimensions, this could have somehow influenced adherence to the CCT with GRADIOR. However, this did not happen, which could be explained because the sample was made up of people with MCI and mild dementia, and not people in advanced stages of the disease.

In this regard, Tolea et al. [56] pointed out that cognitive deterioration is linked with physical deterioration. In this way, people who progress in their cognitive deterioration also progress in their physical deterioration. These changes can also vary according to the etiology, and per example, people with vascular dementia progress more quickly to physical deterioration than people with Alzheimer’s dementia.

Despite our findings, it’s necessary that technologies, including CCT programs, should be available and accessible not only in a clinical center, but also at home in order to improve the access people suffering physical limitations to the treatment [57,58].

### 4.4. Psychological Profile and Adherence

None of the psychological variables (expectations, motivation, and mood) were associated with the CCT adherence. Regarding the initial expectations, the participants agreed that the CCT with GRADIOR would help to improve their memory, quality of life, free time and relating to others. However, our findings did not point to any of these expectations as predictors of adherence. Probably, GRADIOR could help to satisfy these expectations and consequently 62.8% of the participants were adherents (compliant and persistent).

In fact, the users of the “GRADIOR” CCT program improved their social network throughout the sessions. They found it easier to meet and interact with new people because they shared the same needs and interests *“now, I have friends who understand me, and I can have a coffee with them after working with GRADIOR”.* According to the systematic review of Heins et al. [59], interventions with technology have shown to improve the social support network and therefore alleviate feelings of loneliness. Similarly, CCT probably encourages PwD to become active and participate in the group, improving their mood and motivation [38,60,61]. In our study, mood did not help to predict adherent.

Regarding the level of motivation, it did not help to predict adherent. However, we observed that participants did not need external motivation to participate in the RCT. It can be explained by the consent granted by each participant, since this could be a bias of their participation and voluntary commitment.

The questionnaire used to measure expectations, motivation and mood could have been influenced in part by the cognitive impairment in language comprehension commonly associated with people with dementia [62]. Impairment of this cognitive process is usually more evident and significant as dementia progresses, and based on our initial assessment, most people understood the information. Likewise, and with the aim of reducing this problem, the evaluator explained and repeated each of the questions to the participants as many times as necessary. Due to the above, we consider that we tried to reduce biases regarding the understanding of the questions that made up the questionnaire.

### 4.5. Previous Use of Technology and Adherence

The previous use of computers was not associated with higher probability of initiating CCT, contrary to what was mentioned by Turunen, Hokkanen, Bäckman, Stigsdotter-Neely, Hänninen, Paajanen, Soininen, Kivipelto and Ngandu [14]. Currently, the use of technology is probably more frequent, and it is supposed to rise in the next years. Consequently, no previous experience can be a challenge for the patient, and the excitement of using technology for the first time can balance prior experience with it.

Authors should discuss the results and how they can be interpreted from the perspective of previous studies and of the working hypotheses. The findings and their implications should be discussed in the broadest context possible. Future research directions may also be highlighted.

### 4.6. Strengths and Limitations

Regarding the limitations of the study, we are aware that the sample was medium. Nevertheless, this problem is representative of most of the studies carried out with CCT [63]. The main reason was the impact of the onset of a COVID-19 pandemic, which led to the end of the selection process and, in general, the RCT, due to wide mobility restrictions and strict confinement for this vulnerable population. However, this RCT methodologically provides the inclusion of older people with cognitive impairment and their participation for 4 months in a CCT, which is important because most of the published studies on CCT in older people contemplate short periods of CCT [64,65,66]. It makes this study as a good contribution to the clinic field.

Carrying out this study as part of an RCT presented a series of drawbacks in (1) the process of selecting the sample of older people and (2) its maintenance throughout the 4-month CCT. The above is due to different characteristics, which could negatively influence these two scenarios, such as the following: high mortality rate of the sample, mobility alterations, emotional alterations, the stigma of the disease, the start of a pandemic, little financing, and the high cost of resources [21].

On the other hand, we consider that 83.3% to be a high adherence rate for an RCT with the characteristics described and contributes to a little-explored field “adherence to CCT programs”. Even so, the results are more likely to be generalized to people with MCI, as only 37.2% of our sample were people with mild dementia. We consider that adherence to psychosocial approaches will be higher in the early stages of dementia and, consequently, the timely diagnosis of MCI and dementia is strategic for improving the prognosis, quality of life and social health of these people. Indeed, the progression of cognitive impairment leads to fewer resources and less motivation to attend a complex CCT program, even when improvement is predicted [67].

Our study conducted a comprehensive compilation of possible predictors at the person level. Perhaps a variable at the personal level that was not considered in our study was genetics; for example, we know that the APOE-e4 gene is associated with a higher risk of Alzheimer dementia. These types of variables probably exert some influence on the adherence of these people to various treatments, including CCT. Therefore, it would be interesting to develop this type of study.

Orrell, Yates, Leung, Kang, Hoare, Whitaker, Burns, Knapp, Leroi, Moniz-Cook, Pearson, Simpson, Spector, Roberts, Russell, de Waal, Woods and Orgeta [19] suggested contextual variables and person features modify adherence. However, it was not found in our study, but undoubtedly the contexts in which the interventions were carried out could be different and people supporting the intervention could also have different levels of motivation, experience, and skills. All of them could have influenced in the adherence of the participants to the CCT with “GRADIOR”. In any case, it should be noted that all the professionals involved received training and support during the entire RCT to maximize adherence.

Our study did not consider usability and user-experience variables associated with the interface of the “GRADIOR” program in adherence because we had previously performed several usability studies [37] and user-experience. Irazoki, Sánchez-Gómez, Contreras-Somoza, Toribio-Guzmán, Martín-Cilleros, Verdugo-Castro, Jenaro-Río and Franco-Martín [39] and Santos Golino and Flores-Mendoza [68] found that the improvement of instructions helped to rise adherence rates in a CCT program for the elderly. Therefore, it would be interesting for future studies to investigate the association between variables such as usability, user-experience and adherence to CCT programs.

### 4.7. Recommendations and Implications

Our study has a scientific value because it provides a predictive model of adherence based on evidence, indicating several predictive cognitive processes for being considered in the implementation of CCT programs in people with MCI and mild dementia. Adherence to CCT programs is a subject little studied, and most scientific efforts are focused on effectiveness studies [69,70,71,72]. In turn, adherence is a factor that helps determine the effectiveness of a CCT program. Consequently, adherence studies are of great interest, and therefore, scientific efforts should also focus on the development of future research that investigates, corroborates and improves the adherent profiles [73].

The lack of and poor adherence to different psychosocial interventions, including CCT programs such as GRADIOR has clinical implications for people with MCI and mild dementia because it could worsen their cognitive impairment and even social and emotional level. Therefore, this study has clinical value because it helps to propose some strategies to increase adherence to treatment:Make an adequate neuropsychological evaluation, focused on processes such as the following: attention, WM, numerical reasoning, phonological verbal fluency, and cognitive flexibility. To identify those people with the greatest commitment of these processes and, therefore, carry out a more personalized accompaniment and increase the probability of adherence to the CCT.Design personalized CCT plans focused on tasks that involve executive functioning training, specifically in attention, WM, number reasoning, phonological verbal fluency, and cognitive flexibility. Additionally, in turn, modify the level of difficulty of each of the tasks associated with each of the cognitive processes, considering the level and cognitive profile of each of the patients. This will prevent the person from getting bored by the ease of the task or frustrated by its difficulty [74]. In this way, it will be possible to increase the degree of adherence.

It would be interesting for future studies to investigate the influence of involving people with cognitive impairment in decision making on their voluntary participation in the GRADIOR CCT and adherence. Decision making is a process that requires functions such as attention and WM, linked to the adherent rate. We know that decision making could be altered, so the inclusion and support of a reference family member in the process was crucial. However, this point deserves to be studied.

Finally, our study has a practical value because it suggests the importance of developing studies that applying user-centered methodological designs for the development of CCT programs, including end users from the initial stages of their development [74]. It will permit to develop of programs that to meet the physical, cognitive, psychological, and social needs and allow a greater adherence of users. 

## 5. Conclusions

Finally, 62.8% of the participants were adherents to the “GRADIOR” CCT program. Likewise, this group had a high adherence rate of 83.3%. Regarding the predictors, the adherence model consisted of three tests (Digit Symbol of WAIS-III, Arithmetic of WAIS-III and LVF-R). This means that good executive functioning associated with attention, WM, numerical reasoning, phonological verbal fluency, and cognitive flexibility helped predict adherence. Thus, people with MCI and mild dementia with worse scores in these cognitive functions should be considered with higher priority to intervene for preventing the drop out from a CCT program. The adherent group performed better than the non-adherent group in these functions.

## Figures and Tables

**Figure 1 jcm-11-01714-f001:**
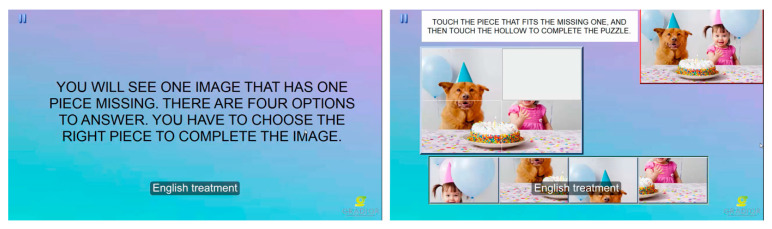
Puzzle exercise. Cognitive modality: executive function. II; software pause.

**Figure 2 jcm-11-01714-f002:**
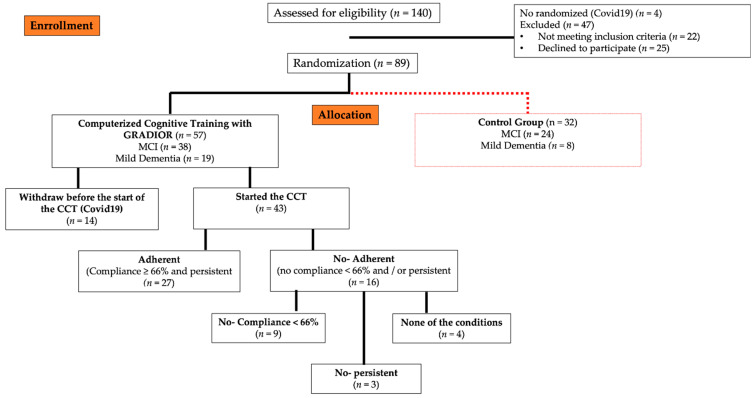
Sample randomization process of randomized clinical trial (RCT). Additionally, conformation of the adherent and non-adherent group in the experimental group (EG). CCT, Computerized cognitive training.

**Table 1 jcm-11-01714-t001:** “GRADIOR” computer-based cognitive training (CCT) plan according to modalities and sub-modalities for participants with mild cognitive impairment (MCI) and mild dementia.

Cognitive Function	Mild Dementia	Both	Mild Cognitive Impairment (MCI)
Orientation		Orientation	
Attention		Selective sequential visual	
		Selective visual-simultaneous	
		Vigilance color	
Memory	Span numbers direct	Hearing short term	Word-Word Associative
		Immediate graphic	Associative image-word
		Span numbers inverse	Span direct lyrics
		Location	
		Verbal compound short term	
		Associative face-name	
		Span direct objects	
Executive Function		Puzzles	Numbers and letters
		Keys	Change rules
		Visual inhibition	Ordination stories
		Interference	
Perception	Visual sizes	Graphic colors	Visual figures
	Visual faces	Text colors	
Calculation		Number identification	
		Arithmetic problems	
Reasoning	Sort charts		

**Table 2 jcm-11-01714-t002:** Baseline characteristics for the adherent and non-adherent group.

Variable	Sub-Categories	T.Student/Mann–Whitney/Χ^2^	*p*	*d-cohen/r_b_*	Adherent			No-Adherent		
					$\bar{x}$ ± SD	Number	%	$\bar{x}$ ± SD	Number	%
Age		−1.185	0.243	−0.374	73.6 ± 6.0			76.1 ± 7.5		
Sex										
	Female	0.044	0.834	0.032		16	61,50%		10	38,50%
	Male					11	64,70%		6	35,30%
Years of education		264.000	0.120	0.222	9.6 ± 2.8			8.4 ± 1.1		
Clinical Group										
	MCI					16	59,30%		11	40,70%
	Mild dementia					11	68,80%		5	31,30%
Adherence Rate					83.3 ± 8.6			59.2 ± 16.1		
MMSE					24.4 ± 2.4			22.6 ± 3.9		
ADAS-Cog: Memory of words					6.1 ± 1.3			6.4 ± 1.5		
ADAS-Cog: word recognition					3.4 ± 1.9			4.6 ± 3.6		
ADAS-Cog: Total					13.7 ± 5.0			17.2 ± 6.7		
TMTA_Mistakes					0.4 ± 0.8			0.4 ± 0.7		
TMTA_Time					12.8 ± 12.0			6.3 ± 2.4		
WAIS-III: Total Digit					10.8 ± 2.6			9.3 ± 2.5		
CAMCOG: Visual Reasoning					2.4 ± 1.4			1.9 ± 1.5		
RBMT: Drawing recognition					7.7 ± 2.2			8.0 ± 2.7		
WAIS-III: Digit Symbol		312.000	0.016 *	**0.444**	10.2 ± 2.7			8.3 ± 2.8		
WAIS-III: Arithmetic		306.500	0.022 *	**0.419**	10.3 ± 2.9			7.7 ± 3.0		
SVF					7.2 ± 3.0			5.6 ± 2.5		
LVF-P					7.7 ± 3.1			6.3 ± 3.2		
LVF-M					8.1 ± 3.8			6.3 ± 3.5		
LVF-R		2.575	0.014 *	**0.812**	8.8 ± 2.5			6.6 ± 3.0		
GDS					4.1 ± 4.0			4.4 ± 2.9		
Health condition					67.0 ± 21.4			76.3 ± 19.6		
Motivation:Attend	Nothing					20	62,50%		12	37,50%
	Somethings					4	66,70%		2	33,30%
	I’m not sure					1	100,00%		0	0,00%
	Quite a lot					2	50,00%		2	50,00%
Expectations: Memory	Nothing					3	60,00%		2	40,00%
	I’m not sure					3	75,00%		1	25,00%
	Quite a lot					13	56,50%		10	43,50%
	A lot					8	72,70%		3	27,30%
Expectations: Quality of life	Nothing					1	33,30%		2	66,70%
	I’m not sure					4	100,00%		0	0,00%
	Quite a lot					14	56,00%		11	44,00%
	A lot					8	72,70%		3	27,30%
Expectations: Free time	I’m not sure					1	100,00%		0	0,00%
	Quite a lot					10	66,70%		5	33,30%
	A lot					16	59,30%		11	40,70%
Expectations: Relating	Nothing					2	100,00%		0	0,00%
	Somethings					2	50,00%		2	50,00%
	I’m not sure					3	100,00%		0	0,00%
	Quite a lot					10	47,60%		11	52,40%
	A lot					10	76,90%		3	23,10%
EQ-5D-5L: Mobility	I have no problem					21	70,00%		9	30,00%
	Minor problems					3	60,00%		2	40,00%
	Moderate problems					2	40,00%		3	60,00%
	serious problems					1	33,30%		2	66,70%
EQ-5D-5L: Self-care	I have no problem					23	63,90%		13	36,10%
	Minor problems					1	33,30%		2	66,70%
	Moderate problems					3	100,00%		0	0,00%
	serious problems					0	0,00%		1	100,00%
EQ-5D-5L: Everyday activities	I have no problem					20	60,60%		13	39,40%
	Minor problems					1	33,30%		2	66,70%
	Moderate problems					5	83,30%		1	16,70%
	serious problems					1	100,00%		0	0,00%
EQ-5D-5L: Pain/discomfort	I have no problem					15	75,00%		5	25,00%
	Minor problems					5	41,70%		7	58,30%
	Moderate problems					3	60,00%		2	40,00%
	serious problems					4	80,00%		1	20,00%
	I can’t					0	0,00%		1	100,00%
EQ-5D-5L: Anxiety/depression	I have no problem					18	72,00%		7	28,00%
	Minor problems					2	33,30%		4	66,70%
	Moderate problems					4	57,10%		3	42,90%
	serious problems					2	50,00%		2	50,00%
	I can’t					1	100,00%		0	0,00%
Prior use of technology	No					4	66,70%		2	33,30%
	Yes					23	62,20%		14	37,80%

Note: $\bar{x}$, mean; * *p*-value ≤ 0.05; CAMCOG, Cambridge Cognition Examination; EQ-5D-5L, EuroQol; GDS, Geriatric Depression Scale; LVF, Lexical Verbal Fluency (forms P, M, R); MCI, mild cognitive impairment; MMSE, Mini-Mental State Examination; RBMT, The Rivermead Behavioural Memory Test; SD, standard deviation; SVF, Semantic Verbal Fluency; TMT, Trail Making Test; WAIS-III, Wechsler Adult Intelligence Scale. Bold Data; significance effect size.

**Table 3 jcm-11-01714-t003:** Model of adherence to a CCT program “GRADIOR”.

Predictor Variable	McFadden R^2^	*p*-Value	Estimate	Standard Error	OR	z	95% CI
WAIS-III: Digit Symbol	0.174	0.019	0.064	0.164	1.066	0.387	0.773–1.470
WAIS III: Arithmetic			0.203	0.153	1.225	1.329	0.908–1.653
LVF-R			0.200	0.146	1.222	1.368	0.917–1.627

Note: CCT, computer-based cognitive training; CI, confidence interval; LVF-R, Lexical Verbal Fluency; OR, odds ratios; WAIS-III, Wechsler Adult Intelligence Scale.

**Table 4 jcm-11-01714-t004:** Mann–Whitney U test. Comparison between people with MCI and mild dementia in relation to each group (adherent and non-adherent).

Variable	Group	Adherent					No-Adherent				
		Mann–Whitney	*p*-Value	*r_b_*	N	$\bar{x}$ ± SD	Mann–Whitney	*p*-Value	*r_b_*	N	$\bar{x}$ ± SD
WAIS-III: Digit Symbol	Mild Dementia	36.000	0.010 **	**−0.591**	11	8.7 ± 0.7	11.000	0.065	**−0.600**	5	6.3 ± 1.5
	MCI				16	11.3 ± 0.6				11	9.2 ± 0.6
WAIS III: Arithmetic	Mild Dementia	31.500	0.005 **	**−0.642**	11	8.2 ± 0.7	16.000	0.205	**−0.418**	5	6.4 ± 1.4
	MCI				16	11.7 ± 0.6				11	8.3 ± 0.9
LVF-R	Mild Dementia	44.000	0.030 *	**−0.500**	11	7.5 ± 0.8	10.000	0.052 *	**−0.636**	5	4.2 ± 1.6
	MCI				16	9.7 ± 0.5				11	7.7 ± 0.6

Note: * *p*-value ≤ 0.05; ** *p*-value ≤ 0.01; $\bar{x}$, mean; LVF, Lexical Verbal Fluency; *r_b_*, Rank biserial correlation; WAIS-III, Wechsler Adult Intelligence Scale. Bold Data; significance effect size.

**Table 5 jcm-11-01714-t005:** Mann–Whitney U test. Comparison between MCI-adherent vs. MCI-non-adherent and mild dementia-adherent vs. dementia-non-adherent.

Variable	Group	Dementia					MCI				
		Mann–Whitney	*p*	*r_b_*	N	$\bar{x}$ ± SD	Mann–Whitney	*p*	*r_b_*	N	$\bar{x}$ ± SD
WAIS-III: Digit Symbol	No-Adherent	17.500	0.464	−0.271	4	6.6 ± 1.9	52.000	0.137	**−0.358**	9	9.9 ± 0.5
	Adherent				12	8.4 ± 0.7				18	10.7 ± 0.7
WAIS III: Arithmetic	No-Adherent	14.000	0.232	**−0.417**	4	6.0 ± 1.8	52.000	0.139	**−0.358**	9	9.0 ± 0.9
	Adherent				12	8.2 ± 0.6				18	10.9 ± 0.7
LVF-R	No-Adherent	14.000	0.244	**−0.417**	4	4.8 ± 1.9	61.500	0.324	−0.241	9	8.2 ± 0.7
	Adherent				12	7.1 ± 0.9				18	9.2 ± 0.6

Note:
$\bar{x}$, mean; LVF, Lexical Verbal Fluency; *r_b_*, Rank biserial correlation; WAIS-III, Wechsler Adult Intelligence Scale. Bold Data; significance effect size.

## Data Availability

The raw data will be provided by the main author to whom it is requested.

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
