# Peer review of "Determinants of Adherence to a “GRADIOR” Computer-Based Cognitive Training Program in People with Mild Cognitive Impairment (MCI) and Mild Dementia"

_jcm, 2022, doi:10.3390/jcm11061714_

Round 1

Reviewer 1 Report

The ms is interesting and well-written. Nevertheless, there are some points that should be clarified or/and revised in order to be suitable for publication. 

1) As regards the two groups of the study, it would be better to add to the respective table in the Results section the indices of the statistical differences-if any-, of the groups in age, gender or educational level.

2) A useful information which could add as regards the interpretation of the results is related to the n of people diagnosed with each type of dementia who included in each group. Moreover, the qualitative examination of their performance, especially of people diagnosed with vascular and frontotemporal dementia, could give us valuable information regarding the level of their executive functioning. Is it possible their results to affect the results of the group?

3) A main issue as regards the discussion of the neurocognitive findings is that the authors seem to dicuss them only in comparison with previous studies. They should proceed to a deeper discussion related to the potential mechanisms (mainly psychological) that are associated with adherence. For example which part of WM could affect adherence, stotrage or processing, and in what way?

Minor issue: The term sex (socially defined) could be replaced with the term gender.    

Author Response

The ms is interesting and well-written. Nevertheless, there are some points that should be clarified or/and revised in order to be suitable for publication. 

Thank you very much for reviewing our manuscript.

  • As regards the two groups of the study, it would be better to add to the respective table in the Results section the indices of the statistical differences-if any-, of the groups in age, gender or educational level.

Authors: Corrected. See page 7 and line 291-293 and 299.

  • A useful information which could add as regards the interpretation of the results is related to the n of people diagnosed with each type of dementia who included in each group. Moreover, the qualitative examination of their performance, especially of people diagnosed with vascular and frontotemporal dementia, could give us valuable information regarding the level of their executive functioning. Is it possible their results to affect the results of the group?

Authors: See page 5-6 and line 258-265 “Then, we have used a non-parametric analysis for two independent samples (Mann Whitney) due to the size of the sample and because not all variables followed a normal distribution (Shapiro-Wilk). This analysis allowed us 1) to compare the performance between adherent and non-adherent group, 2) to investigate if there were significant differences between people with MCI and mild dementia in relation to each group (adherent and non-adherent) and, 3) to evaluate if there were significant differences between MCI-adherent vs MCI-non-adherent and mild dementia-adherent vs mild dementia-non-adherent with respect to the variables (Digit Symbol and Arithmetic of WAIS-III and LVF-R that made up the adherence model”.

Also, See page 8 and line 315-317. Page 9 (table 4 and 5), line 325-341.

We did not want to do more group subdivisions (e.g. types of dementia) because the groups sizes would be very small and this would not be statistically relevant.

  • A main issue as regards the discussion of the neurocognitive findings is that the authors seem to dicuss them only in comparison with previous studies. They should proceed to a deeper discussion related to the potential mechanisms (mainly psychological) that are associated with adherence. For example which part of WM could affect adherence, stotrage or processing, and in what way?

Authors: Corrected. See page 11 and line 413-425

Minor issue: The term sex (socially defined) could be replaced with the term gender.    

Authors: We understand. We are aware of the difference between sex and gender. But our study took into account sex and not gender.

Reviewer 2 Report

This research described a very novel and comprehensive approach to evaluate the adherent of a computer based cognitive training program called 'GRADIOR'. The entire setting of this study is convincing and systematic. There are only a few points I would like the authors to draw to make this piece of work more stringent.

  1. Please provide more information about the pathological conditions of these patents as the dementia can be caused by different disease conditions(not only limited to Alzheimer), while some of the pathological changes/genetic variants can actually bring high adherent risk (nothing to do with the training program) that might interfere with the final conclusion in this work.
  2. With the fact that these patients have mild dementia, how accurate is the survey regarding to their feelings (in Table.2)? The authors should at least discuss this point.
  3. Need more information to describe the values in Table.3 for better understanding the meanings.
  4. The discussion part is a bit redundant, some of the information can be remove d to introduction and it will be more clear and sharp to focus on discussing about the reliability/draw backs of results and the value of this study.
  5. Please remove the extra dots in front of the 'Make an adequate neuropsychological evaluation,...' on page 12.

Author Response

This research described a very novel and comprehensive approach to evaluate the adherent of a computer based cognitive training program called 'GRADIOR'. The entire setting of this study is convincing and systematic. There are only a few points I would like the authors to draw to make this piece of work more stringent.

Thank you very much. This is a new study with important contributions.

  1. Please provide more information about the pathological conditions of these patents as the dementia can be caused by different disease conditions(not only limited to Alzheimer), while some of the pathological changes/genetic variants can actually bring high adherent risk (nothing to do with the training program) that might interfere with the final conclusion in this work.

Authors: Corrected. See page 13 and line 550-554

2. With the fact that these patients have mild dementia, how accurate is the survey regarding to their feelings (in Table.2)? The authors should at least discuss this point.

Authors: Corrected. See page 12 and line 490-497

3. Need more information to describe the values in Table.3 for better understanding the meanings.

Authors: We have added complementary analyzes (See page 5-6 and line 258-265) and two tables 4 and 5 (See page 8 and line 315-317. Page 9 (table 4 and 5), line 325-341). There have also been changes in the discussion section.

4. The discussion part is a bit redundant, some of the information can be remove d to introduction and it will be more clear and sharp to focus on discussing about the reliability/draw backs of results and the value of this study.

Authors: We have removed redundant information throughout the manuscript (e.g. page 2 and line 88-93) . We have added information to the discussion (page 10-12). In the section of limitations and strengths, you will see the implications of the study (see page 12-13). And in recommendations, you will be able to see the scientific, clinical, and practical value of the study (see page 13-14).

5. Please remove the extra dots in front of the 'Make an adequate neuropsychological evaluation,...' on page 12.

Authors: corrected
